# Comparison of the Immunogenicity and Efficacy of rBCG-EPCP009, BCG Prime-EPCP009 Booster, and EPCP009 Protein Regimens as Tuberculosis Vaccine Candidates

**DOI:** 10.3390/vaccines11121738

**Published:** 2023-11-22

**Authors:** Ruihuan Wang, Xueting Fan, Da Xu, Machao Li, Xiuqin Zhao, Bin Cao, Chengyu Qian, Jinjie Yu, Dan’ang Fang, Yujie Gu, Kanglin Wan, Haican Liu

**Affiliations:** National Key Laboratory of Intelligent Tracking and Forecasting for Infectious Diseases, National Institute for Communicable Disease Control and Prevention, Chinese Center for Disease Control and Prevention, Beijing 102206, China; wangruihuan@icdc.cn (R.W.); fanxueting@icdc.cn (X.F.); xuda@icdc.cn (D.X.); limachao@icdc.cn (M.L.); zhaoxiuqin@icdc.cn (X.Z.); bcao1120@163.com (B.C.); qcy88778676@163.com (C.Q.); yjj11087792@163.com (J.Y.); swoon990801@163.com (D.F.); guyujie2022@163.com (Y.G.)

**Keywords:** prime-protein booster, recombinant BCG, vaccine, tuberculosis

## Abstract

Bacillus Calmette–Guérin (BCG) is the only widely used prophylactic tuberculosis (TB) vaccine that can prevent severe TB in infants. However, it provides poor protection in adults, and therefore, there is ongoing research into new TB vaccines and immunization strategies with more durable immune effects. The recombinant BCG and BCG prime-protein booster are two important vaccine strategies that have recently been developed based on BCG and could improve immune responses. In this study, three immune strategies based on four protective antigens, namely, ESAT-6, CFP-10, nPPE18, and nPstS1, were applied to construct recombinant rBCG-EPCP009, EPCP009 subunit protein, and BCG prime-EPCP009 booster vaccine candidates. The short- and long-term immune effects after vaccination in Balb/c mice were evaluated based on humoral immunity, cellular immunity, and the ability of spleen cells to inhibit in vitro mycobacterial growth. At 8 and 12 weeks after the initial immunization, splenocytes from mice inoculated with the BCG prime-EPCP009 protein booster secreted higher levels of PPD- and EPCP009-specific IFN-γ, IL-2, TNF-α, IL-17, GM-CSF, and IL-12 and had a higher IFN-γ^+^CD4^+^ TEM:IL-2^+^CD8^+^ TCM cell ratio than splenocytes from mice inoculated with the rBCG-EPCP009 and EPCP009 proteins. In addition, the EPCPE009-specific IgG2a/IgG1 ratio was slightly higher in the BCG prime-EPCP009 protein booster group than in the other two groups. The in vitro mycobacterial inhibition assay showed that the splenocytes of mice from the BCG prime-EPCP009 protein booster group exhibited stronger inhibition of *Mycobacterium tuberculosis* (*M. tuberculosis*) growth than the splenocytes of mice from the other two groups. These results indicate that the BCG prime-EPCP009 protein booster exhibited superior immunogenicity and *M. tuberculosis* growth inhibition to the parental BCG, rBCG-EPCP009, and EPCP009 proteins under in vitro conditions. Thus, the BCG prime-EPCP009 protein booster may be important for the development of a more effective adult TB vaccine.

## 1. Introduction

Tuberculosis (TB) is a chronic infectious disease that is mainly caused by *M. tuberculosis*. According to the World Health Organization, 10.6 million new TB cases in humans were reported worldwide in 2021, and this is 4.5% more than the 10.1 million new cases reported in 2020 [1]. In recent years, the spread of multi-drug-resistant *M. tuberculosis* strains has made TB an even bigger threat and challenge to global public health and has made the development of a more effective vaccine an urgent need [2]. As of September 2022, there were 16 TB vaccine candidates in clinical trials, and they are classified as recombinant Bacillus Calmette–Guérin (rBCG) vaccines, *M. tuberculosis* live attenuated vaccines, protein/adjuvant vaccines, virus vector vaccines, and inactivated *M. tuberculosis* vaccines [3]. The components of the vaccines are closely related to their immune strategies, and currently, there are two promising strategies for the development of TB vaccines: ① modified recombinant BCG vaccines, which are mainly administered to infants and are a replacement for BCG; ② subunit protein vaccines, which are mainly administered to adults to enhance the protective effect of BCG after primary immunization [4].

As the only vaccine which can prevent TB at present, BCG has been in use since 1921, with a global vaccination rate of more than 90% in infants [5]. It can prevent tuberculous meningitis, miliary tuberculosis, and other serious forms of tuberculosis and plays an important role in the prevention of TB in infants in countries with a high TB burden [6]. However, the protective effect of BCG decreases from adolescence, and it has almost no protective effect on adults [7,8]. Further, the effectiveness of revaccination of adults with BCG is controversial [7,8]. The protective effect of BCG in adults may be reduced because BCG loses some important immunodominant antigen genes during the long-term passage of the strain. These deletion fragments are referred to as the region of differences (RD) (in comparison to *M. tuberculosis*) and include *esxB* and *esxA*, among other fragments [9]. The construction of recombinant BCG vaccines is based on molecular biology technology and involves knocking out the original gene or inducing overexpression of an immunodominant antigen gene based on the parental BCG gene [10]. This strategy overcomes the deficiency in the immune effect and the limited duration of the effect of the original BCG (thus extending its protective effect on adults), while retaining the advantages of BCG in terms of stimulating immune protection (especially in children) [10]. One such recombinant vaccine is the VPM1002 vaccine, in which the *urease C* gene is deleted and transferred into the *Hly* gene of *Listeria monocytogenes* to reduce the pH of phagocytic lysosomes. This improves the phagocytosing and antigen cross-presenting abilities of antigen-presenting cells. VPM1002 was proved to induce stronger immune response and protection against *M. tuberculosis* and was expected to be a viable alternative to BCG [11]. As a result, the construction of safe and effective recombinant BCG vaccines has become a hot topic in the research on TB vaccines.

The immune efficacy of BCG, which is a live attenuated vaccine, is mainly dependent on its effective components and its ability to replicate and persist in the body [12]. Another reason for the reduced protective effect of BCG could be pre-sensitization to non-tuberculosis Mycobacterium and cross-immunization with BCG, which may inhibit BCG replication in humans and lead to reduced colonization ability [12]. Unlike BCG and other live vaccines, subunit protein vaccines are non-replicative, and their immune effect is mainly dependent on the antigenic component and adjuvant and is not affected by pre-sensitization to *M. tuberculosis* and non-tuberculosis Mycobacterium [13]. The antigen component is clear, safe, and especially suitable for people with HIV infection [14]. Since newborns in most countries are vaccinated with BCG after birth, combining the subunit vaccine with BCG can expand the antigen spectrum and induce different types of immune responses, while prolonging the protective effect of BCG and improving the body’s ability to resist TB infection. At present, most TB subunit vaccines are in the clinical phase of research. For example, MVA85A, H4/IC31, and H56/IC31 have been designed for use in adults who have been vaccinated with BCG [15,16,17]. However, there is still no booster vaccine approved for use. It is speculated that the current booster vaccine may be the cross-immune reaction with BCG, which inhibits the growth of BCG and reduces its immune effect. To avoid the effect of the proteins of the booster immunization on BCG colonization and excessive immune response, the target antigen protein of the booster needs to have high specificity for *M. tuberculosis* and produce no or little cross-reactivity with BCG [18]. Although vaccines with two different components of recombinant BCG and subunit proteins have become the main research focus of TB vaccines in recent years, there are few comparative studies on the different immune strategies. Therefore, it is of great significance to clarify the differences in the immunogenicity of these immunization strategies and the level of protection they offer.

At present, the immunodominant antigens of TB mainly include early secretory antigens, virulence-associated antigens, cell membrane antigens, growth phase antigens, and dormant phase antigens, which are potential targets for TB vaccine development [3]. Culture filtrate proteins (CFPs) are secreted proteins present in the supernatant of *M. tuberculosis* culture which can be recognized by T cells and stimulate the production of protective cytokines such as IFN-γ [19]. For example, the RD region proteins ESAT-6 and CFP-10, which are encoded by *esxA* and *esxB* genes, were widely used in early research on TB vaccines [20]. In addition, the PPE18 protein, which belongs to the PE/PPE gene family, is encoded by the *Rv1196* gene, and is enriched with the N-terminal proline (P) and glutamate (E) sequence, has been shown to stimulate rapid proliferation of human T cells in in vitro experiments and can be used as a potential anti-TB vaccine component [21]. Further, the *Rv0934* gene encodes phosphate-specific transport substrate binding protein-1 (PstS1), which is involved in the active transport of inorganic phosphate across the membrane and can induce activation of mouse CD8^+^ T cells and produce Th1 and Th17 immune-protective responses [22]. Among these, PPE18 and PstS1 are expressed at low levels in BCG.

Due to the complex composition of *M. tuberculosis*, the immunogenicity of individual antigens is low, and it is difficult to provide sufficient immune protection with a single antigen. Therefore, combining multiple immunodominant antigens to construct multi-component vaccines is a popular trend in TB vaccine development. In this study, we selected four immunodominant antigens, namely, ESAT-6, CFP-10, nPPE18 (epitope-enriched antigen), and nPstS1 (epitope-enriched antigen), which had been validated in our previous study [23], to construct recombinant BCG and subunit proteins. First, we induced expression of the fusion protein EPCP009 containing ESAT-6, CFP-10, nPPE18, and nPstS1, and used DDA/poly:IC as an adjuvant to create the EPCP009/DP subunit vaccine. In addition, the genes of the four immunodominant antigens were concatenated and introduced into the pMV361 plasmid, and the recombinant BCG vaccine rBCG-EPCP009 was constructed using the BCG Chinese strain as the parent. With Balb/c mice as a model, we examined the short- and long-term immune responses and immune memory levels, in vitro mycobacterial growth inhibition ability, and safety after immunization with the BCG, rBCG-EPCP009, BCG prime-EPCP009/DP booster, and EPCP009/DP vaccines. We explored the optimal vaccine components and immunization strategies for constructing a novel and effective TB vaccine and provided a theoretical basis for the study of prime–boost immunization strategies.

## 2. Materials and Methods

### 2.1. Bacterial Strains and Culture Conditions

*M. tuberculosis* H37Rv, *M. bovis* BCG China (purchased from China National Institutes Food and Drug Control), and rBCG strains were cultured on Difco™ Middlebrook 7H9 (BD, NJ, USA) or Difco™ Middlebrook 7H10 agar (BD, NJ, USA), supplemented with 10% oleic-albumin-dextrose-catalase (OADC) (BD, NJ, USA), 0.5% glycerol, and 0.05% Tween 80. *Escherichia coli* DH5α, and BL21(DE3) were cultured in Luria-Bertani medium or agar and used for cloning and expression. Kanamycin was used at a concentration of 25 μg/mL and ampicillin used at a concentration of 100 μg/mL.

### 2.2. Recombinant rBCG-EPCP009 and the Fusion Protein EPCP009

The expression plasmid pET43.1a-009 and shuttle plasmid pMV361-009 were constructed by Universal Biosystems Co., Ltd. (Anhui, China). The procedure is briefly described here: EPCP009 (*esxA*, *esxB*, *nPPE18*, and *nPstS1* genes) was attached to the pET43.1a vector and pMV361 vector via a linker. The gene sequence of EPCP009 is depicted in Appendix A. As an integral component of fusion protein recombination, the linker is an amino acid chain that connects two fusion genes and has a certain degree of flexibility that allows the proteins on both sides to perform their independent functions [24]. Non-polar hydrophobic amino acids, such as glycine (Gly) and serine (Ser), are commonly used. In the design of our fusion protein vaccines, the linker we chose was Gly-Gly-Ser-Gly-Gly, which is encoded by the GGTGGTTCTGGCGGT sequence.

The pMV361-009 plasmid was transformed into the BCG receptor state by electroporation (voltage, 2.5 kV; capacitance, 25 μF; resistance, 1000 Ω) and screened with the 7H10 resistance medium containing 25 μg/mL kanamycin. A monoclonal colony of rBCG-EPCP009 was cultured in 7H9 medium containing 25 μg/mL kanamycin. The integration of pMV361-009 into BCG was verified by PCR, and the amplified product was sent to the TsingKe Biotech Corp. (Beijing, China) for sequencing to verify the successful construction of the rBCG-EPCP009 strain. The expression level of the target gene was verified by extracting mRNA and using it for RT-qPCR. The primers for verifying the plasmid integration of rBCG-EPCP009 and mRNA expression levels of rBCG-EPCP009 are shown in Table 1.

The transformation of plasmids and the expression and purification of EPCP009 protein are described in previous articles [25]. Briefly, pET43.1a-009 was transformed into *E. coli* BL21 (DE3). A monoclonal colony of BL21 was cultured in Luria-Bertani liquid medium containing 100 μg/mL ampicillin (37 °C, 180 rpm). The bacterial cultures were induced with 1 mM isopropyl β-d-1-thiogalactopyranoside for 6 h. Then, the proteins were purified from the inclusion bodies via nickel chelate chromatography (GE Healthcare, Bensalem, PA, USA). Endotoxin was removed from the purified proteins using a ToxinEraser™ Endotoxin Removal Kit (GenScript, Piscataway, NJ, USA), and the proteins were filtered through 0.45-μm filters.

### 2.3. Immunization Regimens and Sample Collection

Specific pathogen-free 4- to 6-week-old female Balb/c mice were obtained from Beijing HFK Bioscience Co., Ltd. (Beijing, China). In order to evaluate the impact of different immunization schemes (shown in Figure 1), the mice were randomly divided into five groups of six mice each for vaccination. Mice of group 1 (PBS) were immunized with 100 μL of phosphate-buffered saline (PBS, pH 7.4); mice of groups 2 (BCG) and 3 (rBCG-EPCP009) were immunized intradermally with 1 × 10^6^ colony-forming units (CFUs) of the BCG/rBCG strain in 100 μL of PBS (pH 7.4); mice of group 4 (BCG + EPCP009) were immunized intradermally with 1 × 10^6^ CFUs of the BCG strain at week 0 and immunized subcutaneously three times (at weeks 0, 2, and 4) with 10 μg EPCP009 protein formulated in DP (composed of 250 μg dimethyldioctadecylammonium [DDA]/50 μg poly I:C) in 100 μL of PBS (pH 7.4); mice of group 5 (EPCP009) were immunized subcutaneously three times (at weeks 0, 2, and 4) with 10 μg EPCP009 protein formulated in DP. Eight or twelve weeks from the first immunization, blood samples were collected from the orbits of the mice and they were sacrificed by cervical dislocation. The coagulated blood was centrifuged at 4000 rpm for 10 min, and the serum was stored at −20 °C for antibody titer analysis. Spleen cells were isolated under aseptic conditions by using mouse lymphocyte separation medium (Dakewe, Beijing, China) according to the manufacturer’s instructions.

Mice in each group were immunized with PBS, BCG, rBCG-EPCP009, BCG+EPCP009/DP, or EPCP009/DP, delivered by either the intradermal or the subcutaneous route. Eight or twelve weeks from the first booster, blood samples (from the orbit) and spleen cells were collected to analyze their immunogenicity and in vitro protection ability.

### 2.4. Luminex Cytokine Test and Enzyme-Linked Immunospot Assay

The nine cytokines (IFN-γ, IL-2, TNF-α, IL-17, GM-CSF, IL-12, IL-4, IL-6, and IL-10) representing different cellular immune types were detected by Luminex assay, and Th1-type cytokine IFN-γ and Th2-type cytokine IL-4 were detected by the Enzyme-linked immunospot (ELISpot) method, as previously described [25]. What is different is that the cells were stimulated with PPD and EPCP009, respectively. Briefly, the mouse splenocytes were plated in 96-well Luminex Mouse Magnetic Assay (R&D Systems, Minneapolis, MN, USA) culture plates at 2 × 10^5^ cells/well and cultured with 10 μg of the PPD or EPCP009 antigen (i.e., 50 μL of a 200 ng/μL solution) for 24 h. We added biotin antibodies and antibodies separately, wash the cells, and then perform testing. In addition, spleen cells detected by the ELISpot method were stimulated with 2 μg of PPD or EPCP009 protein, respectively.

### 2.5. ELISA

The IgG, IgG1, and IgG2a antibody concentrations in the serum were measured by ELISA [25]. Briefly, 96-well ELISA plates were coated with 200 ng of the PPD or EPCP009 protein and incubated overnight at 4 °C and then were blocked with 3% skimmed milk for 2 h at 37 °C. Two-fold dilutions serum of each experimental group was added and incubated for 2 h at 37 °C. We added 100 μL/well HRP-conjugated goat anti-mouse IgG, IgG1, or IgG2a (Sigma, St. Louis, MO, USA) diluted to 1:5000 in PBS. The plates were incubated for 1 h at 37 °C. We added 100 μL tetramethylbenzidine substrate for 15 min at 37 °C. The reactions were terminated with the addition of 100 μL/well 2 M H_2_SO_4_, and absorbance was measured on an ELISA plate reader at a wavelength of 450 nm. Absorbance values of >2.1 (OD450 [experimental group]/OD450 [PBS control]) were positive.

### 2.6. Flow Cytometry

In order to further analyze the immune mechanism of each vaccine, the spleen cells of each group were stimulated with PPD and EPCP009 after 8 and 12 weeks of the initial immunization, and the proportions of CD3^+^CD4^+^Th cells and CD3^+^CD8^+^CTL cells among T cells were analyzed by flow cytometry combined with intracellular cytokine staining. Following this, the expression of CD62L and CD44 was examined by sorting markers for determining the proportion of effector memory T cells (TEM cells: CD62Llo CD44hi) and central memory T cells (TCM cells: CD62Lhi CD44hi) in the CD4^+^ or CD8^+^ cell populations, as well as the proportion of cells secreting IFN-γ among TEM cells and the proportion of cells secreting IL-2 among TCM cells.

The mouse splenocytes were plated in 96-well culture plates containing 100 μL Luria-Bertani culture medium at a density of 2 × 10^7^ cells/well. Then, 2 × 10^6^ cells from the 100 μL cell suspension were added to 96-well culture plates, to which 100 μL of the prepared stimulation mixture was also added to obtain final concentrations of 10 μg/mL of specific antigens, 1 μg/mL of the CD28 antibody, 1 μg/mL of the CD49d antibody, and 1000× dilution of the Brefeldin A (BFA) blocker, and the cultured cells were stimulated for 8 h in a 5% CO_2_ incubator at 37 °C after mixing. Two positive control wells were set up, and the non-specific stimulant Cell Stimulation Cocktail (with BFA) was added to the wells. Lymphocytes were collected from the culture well in a 5 mL flow tube, to which 2 mL of PBS was added and mixed well. The solution was then centrifuged at 300× *g* for 5 min, and the supernatant was discarded. This solution was mixed well with 1 mL of Ghost Dye™ Violet 510 (for labeling dead cells), prepared by diluting 1000 times with PBS, and incubated for 15 min at room temperature in the dark. Next, 2 mL of flow staining wash was added and mixed well. The solution was centrifuged at 300× *g* for 5 min, and the supernatant was discarded. The cell pellet was resuspended in 100 μL of flow cytometry washing solution, and fluorescence-labeled antibodies (CD3, CD4, CD8, CD62L, and CD44) (BioLegend, San Diego, CA, USA) of cell surface antigens were added, mixed well, and incubated for 20 min at room temperature in the dark. Next, 2 mL of flow staining wash was added and mixed well, and the supernatant was discarded after centrifugation at 300× *g* for 5 min. Cell fixative (0.5 mL) was added to each tube, mixed, and incubated for 20 min at room temperature in the dark. The solution was centrifuged at 300× *g* for 5 min, and the supernatant was discarded. Following this, 2 mL of cytoplasmic membrane breaking lotion was added, mixed well, and centrifuged at 300× *g* for 5 min. The supernatant was discarded, and the pellet was mixed with a solution containing fluorescence-labeled antibodies (IL-2, TNF-α, and IFN-γ) (BioLegend, CA, USA) against the cytoplasmic antigens, mixed well, and incubated in the dark at room temperature for 30 min. Then, 2 mL of cytoplasmic membrane-breaking lotion was added to each tube and mixed well. The solution was centrifuged at 300× *g* for 5 min, and the supernatant was discarded. The cells were resuspended in 0.5 mL of cell stain/wash and subjected to flow cytometry analysis (Cytek NL-CLC3000, CA, USA). The data obtained were analyzed with the FACS Diva 8.0 software.

### 2.7. In Vitro Mycobacterial Growth Inhibition Assay

The mycobacterial growth inhibition assay (MGIA) could evaluate the ability of the splenocytes of mice to inhibit *M. tuberculosis* growth under in vitro conditions and evaluate the protective effect of TB vaccines, which is in line with the 3R principles in animal experiments: reduction, replacement, refinement. The protocol was adapted from a previously published MGIA protocol for mice splenocytes and further optimized in this laboratory [26]. Briefly, the cells were adjusted to a density of 2 × 10^6^ splenocytes per 300 µL and *M. tuberculosis* H37Rv was diluted to a concentration of 50 CFUs/300 µL. We added bacterial aliquots (300 µL) to the splenocytes to form a splenocyte–mycobacteria co-culture (600 µL), and this was incubated at 37 °C for 4 days in 24-well plates. Subsequently, the co-cultures were collected and sterile water (500 µL) was added to each tube and mixed. A total of 50 μL of the solution was inoculated on 7H10 medium containing 5 μg/mL thiophene 2-carboxylic acid hydrazine (TCH) and 10% OADC supplement and incubated at 37 °C for 3 weeks. The number of bacteria was counted, and the data were presented as the log_10_CFU value of the total number per sample.

### 2.8. Statistical Analysis

The experimental data were expressed as the mean ± SD using GraphPad Prism 8. One-way analysis of variance (ANOVA) with Tukey’s multiple comparison tests were used for comparison of more than two groups. *p*-values < 0.05 were considered to indicate statistical significance.

## 3. Results

### 3.1. Verification of rBCG-EPCP009 and the Fusion Protein EPCP009

The whole genome of rBCG-EPCP009 was extracted with the cetyltrimethylammonium bromide (CTAB) method and was amplified by PCR. The PCR-amplified bands were verified by agarose gel electrophoresis, and the sequencing results were consistent with expectations (Figure 2B). RT-qPCR confirmed that the levels of the antigen-specific mRNA of nPPE18 and nPstS1 in rBCG-EPCP009 were higher than those in BCG. The mRNA levels of nPPE18 and nPstS1 were five to ten times higher in the rBCG-EPCP009 group than in BCG (Figure 2C). The expression plasmid pET43.1a-009 was transformed into *E. coli* BL21, and the predicted molecular weight of the expressed EPCP009 protein was 51 kD. The recovery yield of EPCP009 proteins was around 0.8 mg/L of medium. SDS-PAGE indicated that the molecular weight of the proteins was as per expectations, and the purity level of the proteins was high (Figure 2D).

### 3.2. Long-Term Induction of High Levels of Multiple Protective Cytokines by BCG+EPCP009

To better compare the short- and long-term cellular immunity levels and the types of immune responses induced in mice after immunization with the different strategies, the spleens of the immunized mice were isolated under aseptic conditions at 8 and 12 weeks from the first immunization and stimulated with 10 μg each of the PPD and EPCP009 antigens. The levels of nine cytokines, such as Th1-, Th2-, and Th17-type cytokines and other types of cytokines, were detected in culture supernatants with the Luminex kit.

Eight weeks after the initial immunization, mouse spleen lymphocytes were stimulated by PPD (Figure 3). For all of the cytokines compared, the PBS group secreted lower levels of cytokines than the other groups. Compared with the BCG group mice, the rBCG-EPCP009 and BCG+EPCP009 group of mice produced significantly higher levels of IFN-γ, IL-2, TNF-α, IL-17, GM-CSF, IL-12, IL-4, and IL-6 (*p* < 0.05). The BCG+EPCP009 group of mice produced significantly higher levels of IFN-γ, IL-2, TNF-α, IL-17, GM-CSF, IL-12, IL-4, and IL-6 than the rBCG-EPCP009 group (*p* < 0.05) and EPCP009 group of mice (*p* < 0.05). The level of the Th2-type cytokine IL-4 was lower than 100 pg/mL in the BCG, rBCG-EPCP009, BCG+EPCP009, and EPCP009 groups.

Eight weeks after the initial immunization, mouse spleen lymphocytes were stimulated with the EPCP009 protein (Appendix A). As observed for PPD, the PBS group of mice secreted lower levels of cytokines than mice from the other groups. Compared with the BCG group of mice, the rBCG-EPCP009, BCG+EPCP009, and EPCP009 groups of mice produced higher levels of IFN-γ, IL-2, TNF-α, IL-17, GM-CSF, IL-12, and IL-6 (*p* < 0.05). Further, the BCG+EPCP009 group of mice produced significantly higher levels of IFN-γ, IL-2, TNF-α, IL-17, GM-CSF, and IL-12 than the rBCG-EPCP009 and EPCP009 groups of mice (*p* < 0.05).

Twelve weeks after the initial immunization, mouse spleen lymphocytes were stimulated by PPD (Figure 4). The PBS group of mice secreted low levels of cytokines. Compared with the BCG group of mice, the rBCG-EPCP009 and BCG+EPCP009 groups of mice produced significantly higher levels of IFN-γ, IL-2, TNF- α, IL-17, GM-CSF, and IL-12 (*p* < 0.05); further, the BCG+EPCP009 group of mice produced significantly higher levels of IFN-γ, IL-2, TNF-α, IL-17, GM-CSF, and IL-6 than the rBCG-EPCP009 and EPCP009 groups of mice (*p* < 0.05). In conclusion, the BCG+EPCP009 group of mice produced higher levels of PPD-specific cytokines, with a bias toward Th1- and Th17-type cytokines, while the BCG group of mice did not produce IL-17 or GM-CSF.

Twelve weeks after the initial immunization, mouse spleen lymphocytes were stimulated with the EPCP009 protein (Appendix A). The PBS and BCG groups of mice secreted low levels of the EPCP009-specific cytokines. The BCG+EPCP009 group of mice produced significantly higher levels of IFN-γ, IL-2, TNF-α, IL-17, GM-CSF, and IL-6 than the rBCG-EPCP009 and EPCP009 groups of mice (*p* < 0.05). The levels of IL-4 and IL-10 produced by mice from all of the groups were lower than 100 ng/μL.

### 3.3. Long-Term Induction of High Levels of IFN-γ in the Spleen of Mice Vaccinated with BCG+EPCP009

At 8 and 12 weeks after the first immunization, spleens were isolated under aseptic conditions and stimulated with 2 μg PPD and 2 μg EPCP009, and the culture supernatant was assayed for the Th1-type cytokine IFN-γ and the Th2-type cytokine IL-4 using the ELISpot kit.

Eight weeks after the initial immunization, compared with the PBS group of mice, the BCG, rBCG-EPCP009, and BCG+EPCP009 groups of mice produced higher levels of PPD-specific IFN-γ and IL-4 (*p* < 0.05), and the rBCG-EPCP009, BCG+EPCP009, and EPCP009 groups of mice produced higher levels of EPCP009-specific IFN-γ (*p* < 0.05, Figure 5). Compared with the BCG group of mice, both the rBCG-EPCP009 and BCG+EPCP009 groups of mice produced higher levels of IFN-γ in response to PPD or EPCP009 stimulation (*p* < 0.01). However, the BCG+EPCP009 group of mice produced significantly higher levels of IFN-γ than the rBCG-EPCP009 and EPCP009 groups of mice (*p* < 0.001).

At 12 weeks after immunization, the PBS group still secreted lower levels of IFN-γ and IL-4 cytokines than the four immunization groups. Compared with the PBS group of mice, the BCG, rBCG-EPCP009, and BCG+EPCP009 groups of mice produced higher levels of PPD-specific IFN-γ and IL-4 (*p* < 0.05) (Figure 5); the BCG+EPCP009 and EPCP009 groups of mice produced higher levels of EPCP009-specific IFN-γ (*p* < 0.05) (Figure 5). The BCG+EPCP009 group of mice produced significantly higher levels of PPD- and EPCP009-specific IFN-γ than the BCG, rBCG-EPCP009, and EPCP009 groups of mice (*p* < 0.01) (Figure 5).

### 3.4. Induction of Th1-Specific Antibody Types by BCG+EPCP009

Total IgG, IgG1, and IgG2a antibody titers and IgG2a/IgG1 ratios were compared between the PBS group (control group) and the four immunization groups at 8 and 12 weeks after immunization. The *p*-values indicating the significance of the differences were determined by one-way ANOVA with Tukey’s multiple comparison tests.

At 8 and 12 weeks after immunization, the levels of PPD- and EPCP009-specific IgG, IgG1, and IgG2a antibodies in serum samples from the mice were detected by indirect ELISA, and the immune response type (Th1/Th2) was determined by calculating the ratio of IgG2a to IgG1.

With regard to the levels of PPD-specific antibodies (Figure 6), at 8 and 12 weeks after immunization, compared with the PBS group of mice, the BCG, rBCG-EPCP009, and BCG+EPCP009 groups of mice produced higher levels of IgG, IgG1, and IgG2a antibodies, respectively. At 8 weeks, the BCG+EPCP009 group of mice produced higher levels of IgG, IgG1, and IgG2a antibodies than the BCG group of mice and higher levels of IgG1 and IgG2a antibodies than the rBCG-EPCP009 group of mice (*p* < 0.05). The IgG2a/IgG1 ratio of the BCG, rBCG-EPCP009, and BCG+EPCP009 groups was greater than 1, and there was no statistically significant difference between the groups.

With regard to the levels of EPCP009-specific antibodies (Figure 6), after 8 weeks, the BCG+EPCP009 and EPCP009 groups of mice produced higher levels of IgG, IgG1, and IgG2a antibodies than the rBCG-EPCP009 group of mice (*p* < 0.05). At 12 weeks after immunization, compared with the rBCG-EPCP009 group of mice, the BCG+EPCP009 group of mice produced higher levels of IgG and IgG2a antibodies (*p* < 0.05), and the IgG2a/IgG1 ratio of the BCG+EPCP009 group was slightly higher than that of the other three immunization groups.

### 3.5. Consistently High Levels of IFN-γ^+^TEM and IL-2^+^TCM Cells in the Spleens of Mice Vaccinated with BCG+EPCP009

Eight weeks after immunization, the proportions of PPD- and EPCP009-specific IFN-γ^+^CD4^+^TEM and IFN-γ^+^CD8^+^TEM cells were significantly increased in the BCG, rBCG-EPCP009, BCG+EPCP009, and EPCP009 groups compared with the PBS group (*p* < 0.001) (Figure 7A,B). Further, the proportions of PPD- and EPCP009-specific IL-2^+^CD4^+^TCM and IL-2^+^CD8^+^TCM cells in the rBCG-EPCP009 and BCG+EPCP009 groups were significantly increased in comparison to the PBS group (*p* < 0.001). In contrast, the BCG and EPCP009 groups of mice did not produce IL-2^+^CD4^+^TCM or IL-2^+^CD8^+^TCM cells (Figure 7C,D). Twelve weeks after immunization, the BCG group of mice did not produce IFN-γ^+^TEM or IL-2^+^TCM cells, while the rBCG-EPCP009 and BCG+EPCP009 groups produced PPD- and EPCP009-specific IFN-γ^+^CD4^+^TEM, IFN-γ^+^CD8^+^TEM, IL-2^+^CD4^+^TCM, and IL-2^+^CD8^+^TCM cells (*p* < 0.05).

The BCG+EPCP009 group of mice produced significantly higher proportions of PPD- and EPCP009-specific IFN-γ^+^CD4^+^TEM and IL-2^+^CD8^+^TCM cells than the rBCG-EPCP009 and EPCP009 groups of mice when the three experimental immunization groups were compared (*p* < 0.05). In addition, the proportions of PPD- and EPCP009-specific IFN-γ^+^CD8^+^TEM and IL-2^+^CD4^+^TCM cells in the BCG+EPCP009 group were equivalent to those of the rBCG-EPCP009 group.

### 3.6. Better Growth Inhibition of H37Rv by the BCG Prime-EPCP009 Booster Than by rBCG-EPCP009 and EPCP009

To determine the ability of immunized mice to inhibit *M. tuberculosis* growth under in vitro conditions and the duration of this effect, splenocytes were isolated at 8 and 12 weeks after immunization and co-cultured with H37Rv, and the number of colonies formed were counted. At 8 weeks after immunization, the highest colony counts were observed in the PBS-negative control group. Compared with the PBS group, the colony counts in the BCG, rBCG-EPCP009, BCG+EPCP009, and EPCP009 groups were significantly lower (*p* < 0.01). Compared with the BCG group, the colony counts in the rBCG-EPCP009 and BCG+EPCP009 groups were significantly lower (*p* < 0.01). Further, the BCG+EPCP009 group had the lowest colony count, and this was significantly lower than that of the rBCG-EPCP009 (*p* < 0.001) and EPCP009 groups (*p* < 0.05) (Figure 8).

At 12 weeks after immunization, the PBS-negative control group still had the highest colony count, and the BCG, rBCG-EPCP009, BCG+EPCP009, and EPCP009 groups had significantly lower colony counts than the PBS group (*p* < 0.01). The number of colonies formed was significantly lower in the BCG+EPCP009 group than in the BCG group (*p* < 0.01), but there was no significant difference between the rBCG-EPCP009 and EPCP009 groups. Moreover, the colony counts in the BCG+EPCP009 group were significantly lower than those in the rBCG-EPCP009 (*p* < 0.001) and EPCP009 groups (*p* < 0.001). In conclusion, at 12 weeks after immunization, the BCG+EPCP009 group showed significantly better inhibition than the BCG, rBCG-EPCP009, and EPCP009 groups (Figure 8).

## 4. Discussion

Tuberculosis remains a serious infectious disease that poses a threat to human health. BCG is the only preventive vaccine currently approved for human use. However, while it is effective in preventing severe TB in infants and children, it has severely reduced protection in adolescents and adults [6]. Currently, 16 novel TB vaccine candidates are in clinical trials worldwide, and recombinant BCG and subunit protein vaccines are two important vaccine strategies. Recombinant BCG is intended for neonatal replacement of BCG, and the subunit protein is mainly used for booster vaccines after primary BCG immunization. Both recombinant BCG and the subunit protein are expected to be useful strategies for the development of new vaccines and for the development of immunization protocols which provide stronger and longer-lasting immunity based on the inherited parental BCG [27]. In this study, we constructed recombinant rBCG-EPCP009 and subunit protein EPCP009 vaccine candidates based on the multi-component immunodominant antigen EPCP009 and evaluated the short- and long-term immunogenicity and in vitro protection offered by three immunization strategies, namely, rBCG-EPCP009, BCG+EPCP009, and EPCP009/DP. Our results showed that the BCG prime-EPCP009 booster provided superior and better in vitro protection than rBCG-EPCP009 and EPCP009/DP, as reflected in the following results. That is, BCG+EPCP009 induced stronger PPD- and EPCP009-specific Th1-type and Th17-type immune responses based on the levels of the cytokines GM-CSF and IL-12, and the IFN-γ^+^CD4^+^TEM and IL-2^+^CD8^+^TCM cell ratios, at both 8 and 12 weeks after immunization, while BCG+EPCP009 induced EPCP009-specific antibodies which were indicative of a bias toward Th1-type immune responses. In addition, immunization with BCG+EPCP009 resulted in stronger in vitro Mycobacterium inhibition by splenocytes and had a longer in vitro protective effect.

The selection of an appropriate immunodominant antigen is an important factor in determining vaccine efficacy, irrespective of whether the strategy involves the recombinant BCG or subunit vaccine. rBCG30, which expresses the Ag85B antigen, was the first vaccine candidate of this type to enter the clinical trial phase and showed superior immune efficacy to the parent BCG in both mice and guinea pigs. However, the phase I clinical trials conducted in the United States in 2004 showed that the immunogenicity of rBCG30 was lower than that of BCG, and therefore, the findings from human trials were not encouraging [28]. Subsequently, the strategy for construction of recombinant BCG vaccines changed from overexpression of a single antigen to simultaneous overexpression of multiple immune dominant fusion antigens in order to further improve the immune effect of recombinant BCG. For example, rBCG::XB, which is in the pre-clinical phase, is a novel recombinant BCG that expresses both the Ag85B and HspX antigens and induces a stronger antigen-specific Th1-type immune response in mice than BCG [29]. In addition, rBCG::685A, which expresses both ESAT-6 and Ag85A, also induced higher levels of PPD or r685A-specific IFN-γ in mice [30]. In this study, based on the ESAT-6 and CFP-10 proteins, the fusion gene EPCP009 was constructed by adding the antigens nPPE18 and nPstS1. Recombinant rBCG-EPCP009 overexpressing the EPCP009 protein was constructed via the pMV361 shuttle plasmid. In addition, the ESAT-6, CFP-10, nPPE18, and nPstS1 genes were arranged in tandem, and the fusion protein EPCP009 and DDA/poly:IC adjuvant were mixed to construct the EPCP009/DP vaccine candidate which was used as a booster vaccine after initial immunization with BCG. Compared with mice immunized with rBCG-EPCP009, mice immunized with the BCG prime-EPCP009 booster strategy secreted higher levels of PPD- and EPCP009-specific IFN-γ, IL-2, TNF-α, IL-17, GM-CSF, and IL-12 at 8 and 12 weeks after immunization. In vitro protection assays showed that splenic lymphocytes from BCG+EPCP009-immunized mice showed superior *M. tuberculosis* growth inhibition than splenocytes from rBCG-EPCP009-immunized mice at both 8 and 12 weeks. This indicates that although mice from all of the immunization groups expressed the fusion protein EPCP009, the immune mechanisms induced differed according to the technology platforms and immunization strategies employed.

Under the stimulation of different antigens, Th0 cells differentiate into different Th1 and Th2 cell subsets, and different subsets of T cells secrete different types of cytokines in response to pathogens [31]. As an intracellular bacterium, *M. tuberculosis* mainly produces IL-12 by inducing its secretion from antigen-presenting cells. These antigen-presenting cells induce Th1 cells to release IFN-γ, IL-2, and TNF-α and activate macrophages and cytotoxic T lymphocytes, which then fulfill their function of *M. tuberculosis* clearance [32]. However, IL-4 induces the polarization of Th0 to the Th2 subset, and the production of IL-4 and IL-10 inhibits the anti-infection process and leads to the dissemination of *M. tuberculosis* [33]. Th17 cells are a unique group of Th cells that mainly secrete IL-17 and GM-CSF, which recruit and activate neutrophils to the site of infection. Th17cells and neutrophils enhance the protective immune response by producing IL-12 and inducing a Th1 response [34]. IL-6 is a pro-inflammatory cytokine that can enhance the immune response by promoting the proliferation of T cells, B cells, and other immune cells [35]. According to the latest evidence, the main effector T cell subsets involved in *M. tuberculosis* clearance are Th1 and Th17 cell subsets [36]. However, certain components of the *M. tuberculosis* cell wall, such as ManLAM, can promote IL-10 production and secretion by inhibiting the production of IFN-γ and TNF-α, thereby affecting antigen presentation or inhibiting the migration of effector T cells to the site of *M. tuberculosis* infection and resulting in immune escape [37]. Compared with BCG, the recombinant BCG vaccines rBCG-EPCP009 and BCG+EPCP009 resulted in the secretion of increased levels of PPD- and EPCP009-specific IFN-γ, IL-2, TNF-α, IL-17, GM-CSF, and IL-12 from mouse splenocytes at 8 and 12 weeks after immunization. These results indicate that rBCG-EPCP009 and BCG+EPCP009 induced a protective immune response which was superior to that of BCG. In addition, through comparing the three EPCP009-based immunization strategies, the level of PPD- and EPCP009-specific IFN-γ, IL-2, TNF-α, IL-17, GM-CSF, and IL-6 stimulated by BCG+EPCP009 was found to be significantly higher than that stimulated by rBCG-EPCP009 and EPCP009 at both 8 and 12 weeks after immunization. This result indicates that the combined strategy involving BCG+EPCP009 promoted not only the immune effect of BCG itself but also the immune response to the EPCP009 protein. Thus, BCG and EPCP009 may have induced a stronger immune effect which was brought about by a mutual adjuvant effect.

Memory T cells are mainly divided into TCM cells and TEM cells [38], and T cells are highly heterogeneous in terms of their immune response. When exposed to antigen stimulation, TCM cells mainly secrete IL-2, rapidly proliferate and differentiate into effector T cells, and produce IFN-γ. TEM cells mainly secrete IFN-γ, which migrates to the site of inflammation and rapidly exerts its immune effects [39]. It has been shown that IFN-γ^+^TEM and IL-2^+^TCM cells are biomarkers of vaccine-induced production of anti-*M. tuberculosis* antibodies and early and persistent infection, respectively [40]. The main reason for the decrease in the protective properties of BCG with time may be that its induced protective immune response decreases with time so that it induces a weaker CD8^+^ immune response in adolescents and adults [41]. The results of the present study show that the proportions of neither IFN-γ^+^TEM nor IL-2^+^TCM cells were significantly higher in the BCG group compared with the PBS group at 12 weeks after immunization. This may partly explain why the protective effect of BCG is only maintained for 10–15 years and highlights the need for a TB vaccine with a longer immune memory. At 8 or 12 weeks after immunization, the rBCG-EPCP009 and BCG+EPCP009 vaccines induced significantly higher proportions of PPD- or EPCP009-specific IFN-γ^+^CD4^+^TEM, IL-2^+^CD4^+^TCM, and IL-2^+^CD8^+^TCM cells than BCG. This finding indicates that immune memory was significantly enhanced with the rBCG-EPCP009 and BCG+EPCP009 vaccine candidates. In addition, the proportion of antigen-specific IL-2^+^CD8^+^TCM cells induced by PPD or EPCP009 was significantly higher in the BCG+EPCP009 group than in the rBCG-EPCP009 and EPCP009 groups. When CD8^+^TCM cells were exposed to the antigen again, they rapidly proliferated and differentiated and produced IL-2, while promoting the cytotoxic functions of effector CD8^+^ T cells, which, in turn, enhanced their anti-infective effects [42,43]. Thus, the stronger protective effect of VPM1002 compared to BCG may be related to its ability to induce high levels of TCM production [11]. In conclusion, several studies have demonstrated the correlation between the proportions of IFN-γ^+^ TEM and IL-2^+^ TCM cells and the protective effect of TB vaccines. With regard to the vaccine candidates evaluated in the present study, BCG+EPCP009 induced higher and longer-lasting levels of IFN-γ^+^CD4^+^ TEM and IL-2^+^CD8^+^ TCM cells. Thus, this immunization strategy may provide a longer-lasting immune protective effect.

MGIA is an effective method to comprehensively evaluate the protective effect of TB vaccines in vitro [26,44]. Parra et al. used mice as a model to test the inhibitory effect of splenocytes from vaccine-immunized mice on *M. tuberculosis* in macrophages, and the results were consistent with the results of challenge protection experiments on animals [45]. The results of this study show that at 8 weeks after the initial immunization, splenocytes from the rBCG-EPCP009- and BCG+EPCP009-immunized groups showed stronger in vitro mycobacterial inhibition than splenocytes from the BCG group; further, EPCP009 group splenocytes showed comparable inhibition to BCG group splenocytes. At 12 weeks after the initial immunization, the BCG+EPCP009 group splenocytes still exhibited better inhibition than the BCG group splenocytes, while the rate of inhibition of the rBCG-EPCP009 and EPCP009 groups splenocytes were not significantly different from that of the BCG group splenocytes. This indicates that rBCG-EPCP009 provided better short-term protection than BCG but did not provide long-term protection. At both 8 and 12 weeks after immunization, BCG+EPCP009 exhibited durable superior bacterial growth inhibition to rBCG-EPCP009 and EPCP009; this may be related to its ability to induce the production of PPD- and EPCP009-specific TNF-α, IFN-γ, IL-12, IL-17, IFN-γ^+^CD4^+^ TEM cells, and IL-2^+^CD8^+^ TCM cells.

In recent years, the protective effect of the BCG prime-BCG homologous enhancement strategy has been under debate, and many studies conducted in different countries, such as Japan and Finland, have shown that BCG revaccination does not improve the protective effect of BCG itself and may even cause pathological damage [46,47]. Currently, the immunization strategy involving primary immunization with BCG and a heterologous boost has become the focus for the development of novel vaccines for TB, and several subunit protein and viral vector vaccines are in the clinical phase of evaluation for their application as booster immunization. Different from the previous selection of Mycobacterium bovis antigens as booster vaccines [48], we selected EPCP009 containing four Mtb-specific antigens as a candidate vaccine, expanding the antigen lineage. The results of this study show that the BCG+EPCP009 immunization group exhibited a more durable protective immune response and in vitro *M. tuberculosis* inhibition. Thus, administering the EPCP009 subunit protein vaccine as a booster immunization vaccine for BCG could enhance the immune protective response of the body. However, the limitation of this study is that it only provides a preliminary comparison of different immunization strategies. Furthermore, we will explore the dose, frequency, and interval of boost immunization.

## 5. Conclusions

In this study, we found that the BCG prime-EPCP009 booster induced mice to secrete higher levels of PPD- and EPCP009-specific IFN-γ, IL-2, TNF-α, IL-17, GM-CSF, and the IFN-γ^+^CD4^+^TEM and IL-2^+^CD8^+^TCM cell ratios than the rBCG-EP009 group and EPCP009f/DP group, at both 8 and 12 weeks after immunization. The combination of BCG and EPCP009 boosted the immune response to BCG and EPCP009, respectively, and both PPD-specific and EPCP009-specific cytokine and T-cell immune responses were significantly enhanced. The in vitro protective effect of the BCG prime-EPCP009 booster immunization strategy was superior to that of the rBCG-EPCP009 and EPCP009 protein vaccines. Overall, a comparison of the different strategies revealed that the BCG prime-EPCP009 protein booster immunization strategy was superior to the rBCG-EPCP009 and EPCP009 strategies and may provide a theoretical basis for future immunization strategies of TB vaccines.

## Figures and Tables

**Figure 1 vaccines-11-01738-f001:**
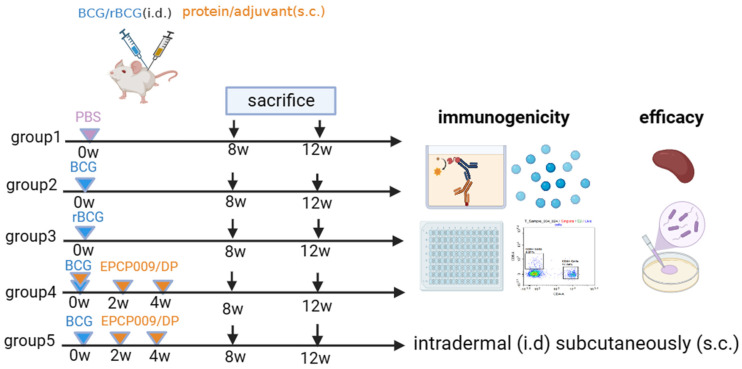
Vaccination regimen schedules.

**Figure 2 vaccines-11-01738-f002:**
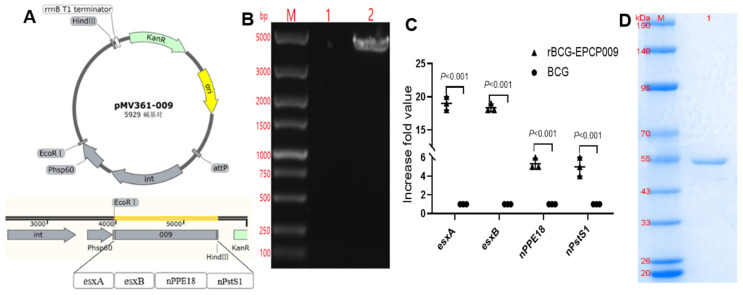
Verification of rBCG-EPCP009 and the fusion protein EPCP009. (**A**) Schematic diagram of the recombinant plasmid pMV361-009. (**B**) Verification of the rBCG-EPCP009 gene. Lane M, standard DNA marker; lane 1, genomic PCR product of BCG; lane 2, genomic PCR product of rBCG-EPCP009. (**C**) mRNA levels of *esxA, esxB, nPPE18,* and *nPstS1* in rBCG-EPCP009, as determined by qRT-PCR. cDNA levels were normalized to the endogenous RNA levels of the internal control 16S rRNA gene. The fold increase in gene expression was calculated using the 2^−ΔΔCT^ method. (**D**) Lane M, molecular weight marker of the pre-stained protein; lane 1, EPCP009.

**Figure 3 vaccines-11-01738-f003:**
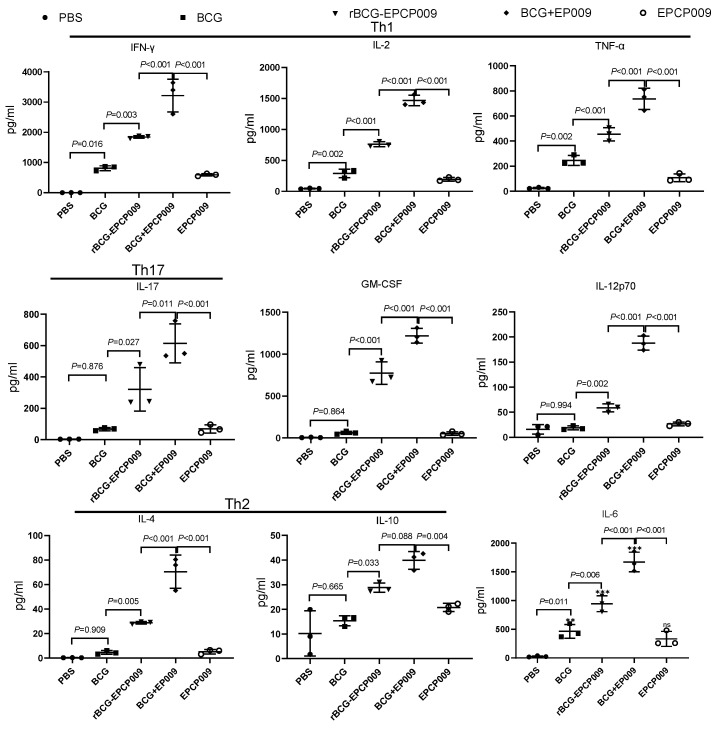
Levels of cytokines secreted by mouse spleen lymphocytes 8 weeks after immunization with the candidate vaccines. Cytokine levels were examined after stimulation with PPD 8 weeks after initial immunization with the different vaccine candidates. The *p* values indicating the significance of differences were determined by one-way ANOVA with Tukey’s multiple comparison tests.

**Figure 4 vaccines-11-01738-f004:**
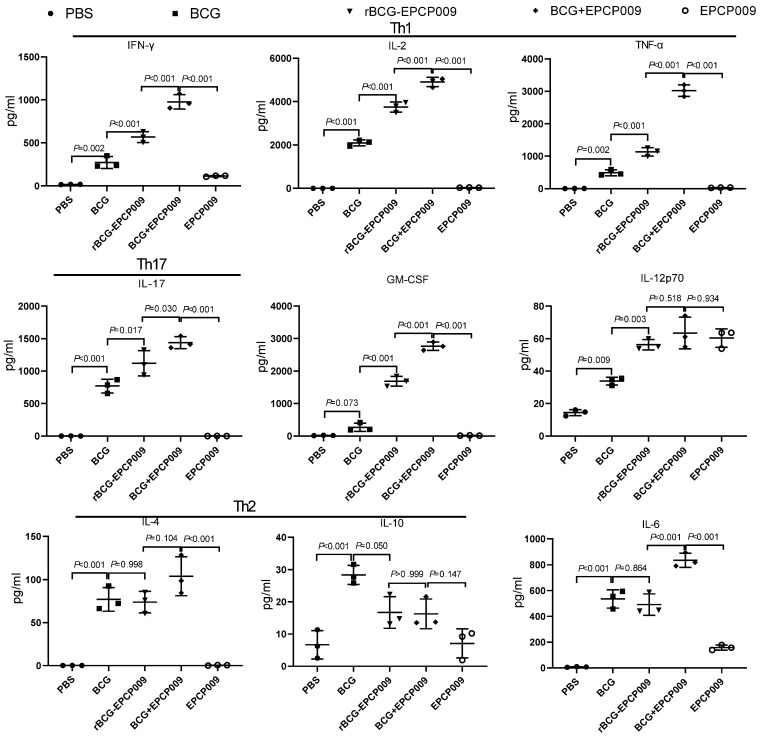
Levels of cytokines secreted by mouse spleen lymphocytes 12 weeks after immunization with the vaccine candidates. Cytokine levels secreted on stimulation with PPD-specific proteins were measured 12 weeks after initial immunization with the different vaccine candidates. The *p*-values indicating the significance of differences were determined by one-way ANOVA with Tukey’s multiple comparison tests.

**Figure 5 vaccines-11-01738-f005:**
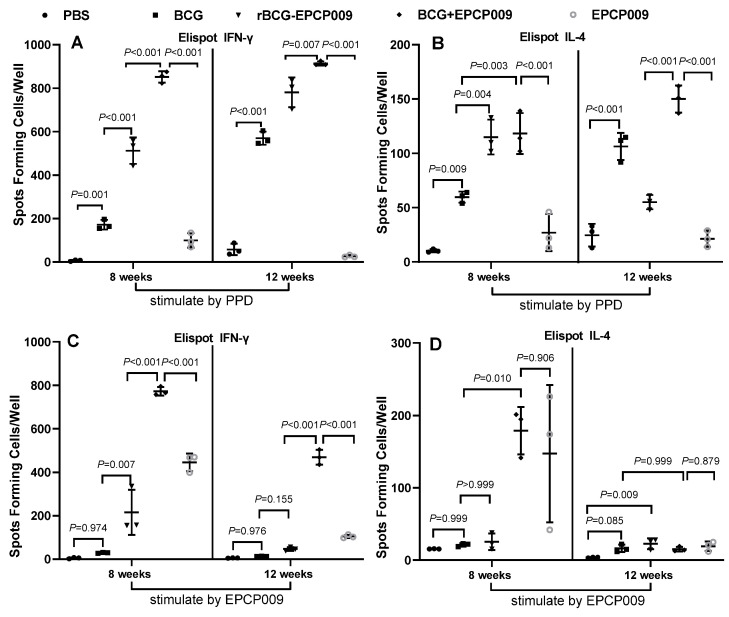
Antigen-specific IFN-γ and IL-4 levels secreted by mouse splenic lymphocytes. The IFN-γ (**A**,**C**) and IL-4 (**B**,**D**) levels were determined at 8 or 12 weeks after the first immunization. The proportion of T cells secreting IFN-γ and IL-4 was determined by the ELISpot assay. The *p* values indicating the significance of the differences were determined by one-way ANOVA with Tukey’s multiple comparison tests.

**Figure 6 vaccines-11-01738-f006:**
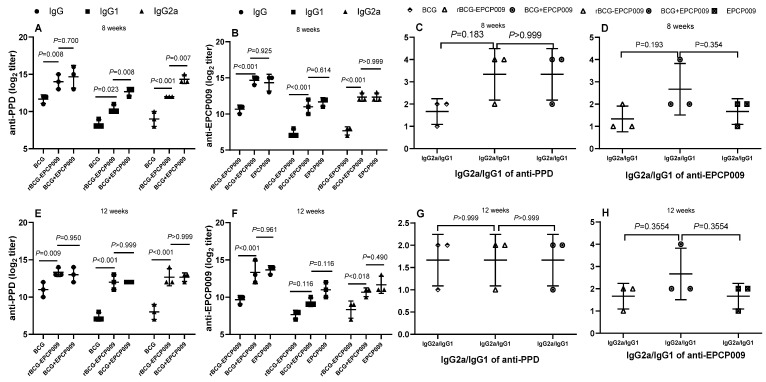
Antibody response of IgG, IgG1, and IgG2a in Balb/c mice immunized with different vaccine candidates at 8 or 12 weeks after the initial immunization. (**A**) PPD-specific and (**B**) EPCP009-specific antibodies titers of total IgG, IgG1, and IgG2a at 8 weeks. (**C**) PPD-specific and (**D**) EPCP009-specific of IgG2a/IgG1 ratio at 8 weeks. (**E**) PPD-specific and (**F**) EPCP009-specific antibodies titers of total IgG, IgG1, and IgG2a at 12 weeks. (**G**) PPD-specific and (**H**) EPCP009-specific of IgG2a/IgG1 ratio at 12 weeks.

**Figure 7 vaccines-11-01738-f007:**
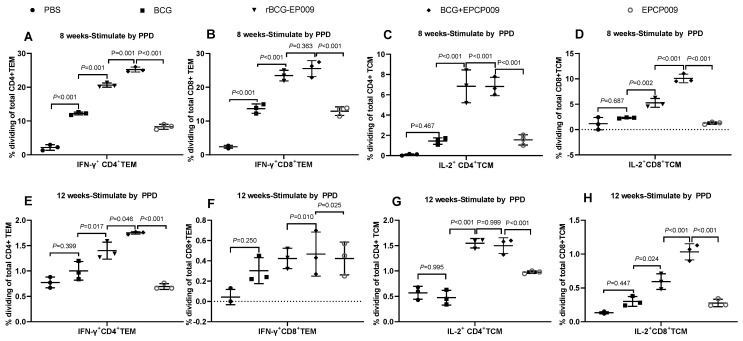
Proportions of various immune cells in spleen cells from immunized mice. Splenocytes from Balb/c mice immunized with the candidate vaccines were stimulated with the PPD antigen at 8 and 12 weeks, and the proportions of IFN-γ^+^CD4^+^TEM (**A**,**E**) and IFN-γ^+^CD8^+^TEM (**B**,**F**) cells and IL-2^+^CD4^+^TCM (**C**,**G**) and IL-2^+^CD8^+^TCM (**D**,**H**) cells were assessed by flow cytometry. The *p*-values indicating the significance of differences were determined by one-way ANOVA with Tukey’s multiple comparison tests.

**Figure 8 vaccines-11-01738-f008:**
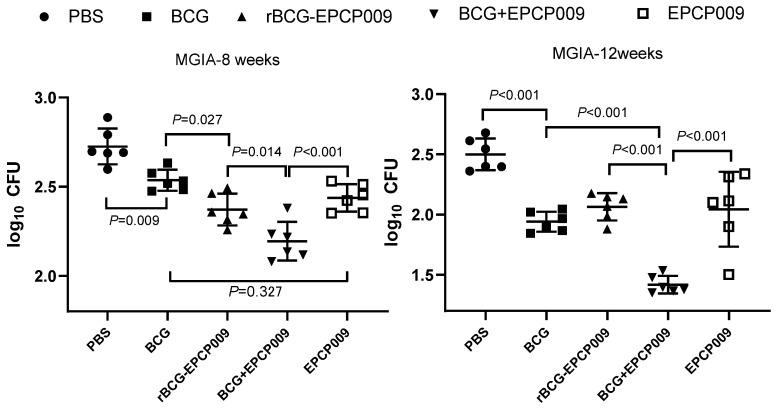
In vitro growth inhibition of *M. tuberculosis* H37Rv with splenocytes isolated from Balb/c mice in different immunized groups. At 8 and 12 weeks after immunization, 2 × 10^6^ splenocytes from immunized mice groups were co-cultured with 50 CFU of *M. tuberculosis*. Splenocytes were obtained from 6 control animals in each group, represented by an individual data point. The *p*-values of the differences were determined by one-way ANOVA with Tukey’s multiple comparison tests.

**Table 1 vaccines-11-01738-t001:** Primers for validation of recombinant BCG.

Primers	Primer Sequences (5′ to 3′)
Integration validation primer F	CGGCTTATCAACTAGATCGGCGCAG
Integration validation primer R	GACGTCAGGTGGCTAGCTGATCA
*esxA* RT-qPCR primer F	TGACAGAGCAGCAGTGGAATTTCG
*esxA* RT-qPCR primer R	CAAGGAGGGAATGAATGGACGTGAC
*esxB* RT-qPCR primer F	AGCCAATAAGCAGAAGCAGGAACTC
*esxB* RT-qPCR primer R	CTAGAAGCCCATTTGCGAGGACAG
*nPPE18* RT-qPCR primer F	TGTCGATGACCAACACCTTGAGC
nPPE18 RT-qPCR primer R	CCAGAACCACCACCCGAAGAAC
nPstS1 RT-qPCR primer F	CGCCTATCTGTCGGAAGGTGATATG
nPstS1 RT-qPCR primer R	GTTGACCTGCTGAGCGGAGATG
16s RT-qPCR primer F	CGCACAAGCGGCGGAGCA
16s RT-qPCR primer R	GCCACAAGGGAACGCCTATCT

## Data Availability

Data are contained within the article.

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
