# Peer review of "Comparison of the Immunogenicity and Efficacy of rBCG-EPCP009, BCG Prime-EPCP009 Booster, and EPCP009 Protein Regimens as Tuberculosis Vaccine Candidates"

_vaccines, 2023, doi:10.3390/vaccines11121738_

Round 1

Reviewer 1 Report

Comments and Suggestions for Authors

Positive feedback. The authors address an important problem. The rationale for looking at the new vaccine candidate is reasonable, and experiments are well presented. I could following the results relatively well (except see comment on figures).

Major comments

I don't think comparing immune response with BCG vs. BCG+subunit is fair as one is single vaccination and another is a double vaccination. Perhaps authors need to have a group with BCG+BCG for best comparison.

Measuring immunity by cytokines without looking at numbers of cells making them seems like an outdated method. Authors should calculate the numbers of Mtb-specific T cells and their phenotype induced by vaccination.

Some data are not well presented, figures are small and text is nearly impossible to read. Figure 2 seems blurry.
Figure 3 could perhaps have only panel A or two panels could be put on top of each other, so the figure is bigger and font is reasonable. The use of barchats is not recommended anymore, please show data from each mouse by dots. Perhaps limits of axes could be made similar in all plots? Fig 3 and 4 show similar data - if you want to address the stability of immunity between 8 and 12 weeks, perhaps you should plot the data as time-plot with 2 time points.
Putting p values is preferred to using stars, indicating actual p value not non-significant values is also recommended.

Figure 8 is important but it is hard to compare the results because y axis limits are different. Also, caption does not provide much info on the experiment done. You need to improve here to show which specific cells contribute to Mtb control in the experiment.

Please show all results for individual gels in supplement and do not use bar charts but show individual values from every gel in Figure 2.

Discussion is not well structured. I do not understand why there is a need to discuss details about Th1/Th2 response, etc. You discussion should be short and to the point. This is a format I follow: 1 - summarize the results, 2 - how these results connect to results of others, 3 - what are limitations of your work, and 4 - what are future directions.

Minor comments

Line 88-90 - what inhibits growth of BCG?

BCG varies dramatically. Can you elaborate where your BCG-China comes from? Some papers could be useful.

Figure 6/7 captions seems too short and not very informative.

Change in format in Figure 5 from vertical to horizontal bar charts makes it harder to interpret the figure.

It is better to have notations at the top, not the bottom (8 weeks/12 weeks in Figure 6).

Line 477: which Figure shows longer immunogeneticity? I did not see that analysis.

Author Response

Positive feedback. The authors address an important problem. The rationale for looking at the new vaccine candidate is reasonable, and experiments are well presented. I could following the results relatively well (except see comment on figures).

Thank you very much for reading and reviewing our manuscript, which helps us to improve it to a better scientifical level. Please find the detailed responses below and the corrections highlighted in the re-submitted manuscript.

Major comments

I don't think comparing immune response with BCG vs. BCG+subunit is fair as one is single vaccination and another is a double vaccination. Perhaps authors need to have a group with BCG+BCG for best comparison.

Response: Thank you for pointing this out. Many past studies have shown that two doses of BCG do not improve the efficacy of BCG. At present, WHO recommends one dose of BCG. Our research focuses on providing novel vaccines with recombinant BCG and BCG+subunit to address the lack of BCG efficacy. In addition, not all BCG+subunits are more effective than BCG, and may even reduce the efficacy of BCG. Therefore, studying the BCG+subunits and comparing with BCG is of great significance. In our next experimental design, we will take your valuable advice and add BCG+BCG group as a control.

Measuring immunity by cytokines without looking at numbers of cells making them seems like an outdated method. Authors should calculate the numbers of Mtb-specific T cells and their phenotype induced by vaccination.

Response: Thank you very much for your opinion. Consistent with your point of view, we also think that it is meaningful to detect Mtb-specific cytokines secreted by T cells, and we presented the Mtb-specific T cells cytokines detected by flow cytometry method in Figure 7. This part is mainly the preliminary presentation of the total 9 cytokines of without using typing assays, and we will consider your comments for improvement in our future work.

Some data are not well presented, figures are small and text is nearly impossible to read. Figure 2 seems blurry.

Figure 3 could perhaps have only panel A or two panels could be put on top of each other, so the figure is bigger and font is reasonable. The use of barchats is not recommended anymore, please show data from each mouse by dots. Perhaps limits of axes could be made similar in all plots? Fig 3 and 4 show similar data - if you want to address the stability of immunity between 8 and 12 weeks, perhaps you should plot the data as time-plot with 2 time points.

Putting p values is preferred to using stars, indicating actual p value not non-significant values is also recommended.

Response: Thank you for your advice. We have changed figures 3 and 4, retaining only panel A, the PPD specific cytokine. Other results were presented in the supplementary figures.  The data of each mouse was represented by dots, and the p-value was the actual value. Figure 3 shows the data at 8 weeks with higher effector cytokine IFN- γ. Figure 4 shows the cytokine levels at 12 weeks, and the cytokines related to memory, such as IL-2, are higher. Cytokines with different functions at different time points varied widely and the y axis were not unified, so they were analyzed in two figures.

Figure 8 is important but it is hard to compare the results because y axis limits are different. Also, caption does not provide much info on the experiment done. You need to improve here to show which specific cells contribute to Mtb control in the experiment.

Response: Thank you for your comments. We have changed y axis limits of figure 8 and updated caption.

Please show all results for individual gels in supplement and do not use bar charts but show individual values from every gel in Figure 2.

Response: Thank you for your comments. We have changed figure 2 to show individual values.

Discussion is not well structured. I do not understand why there is a need to discuss details about Th1/Th2 response, etc. You discussion should be short and to the point. This is a format I follow: 1 - summarize the results, 2 - how these results connect to results of others, 3 - what are limitations of your work, and 4 - what are future directions.

Response: Thank you for your suggestion. We have added the limitations and future directions in the revised manuscript (lines 601-603 and 610-612). Since there is no direct protective marker for TB vaccines, a discussion of the different immune responses induced by vaccine candidates is needed.

Minor comments

Line 88-90 - what inhibits growth of BCG?

Response: Thank you for your advices. We have changed the information in lines 86-89.

BCG varies dramatically. Can you elaborate where your BCG-China comes from? Some papers could be useful.

Response: Thank you for your advices. We have updated the information in lines 132.

Figure 6/7 captions seems too short and not very informative.Change in format in Figure 5 from vertical to horizontal bar charts makes it harder to interpret the figure.

It is better to have notations at the top, not the bottom (8 weeks/12 weeks in Figure 6).

Response: Thank you for your constructive comments. We have changed the captions and charts in figure 6/7 and made the notations at the top (8 weeks/12 weeks in Figure 6).

Line 477: which Figure shows longer immunogeneticity? I did not see that analysis.

Response: Thank you for pointing out the problems. I'm sorry that our description is not very accurate and we have revised the manuscript (line 478).

Kanglin Wan

National Institute for Communicable Disease Control and Prevention, Chinese Center for Disease Control and Prevention, Beijing, China

Email address: wankanglin@icdc.cn

Reviewer 2 Report

Comments and Suggestions for Authors

The manuscript entitled "Comparison of the immunogenicity and efficacy of rBCG-EPCP009, BCG prime-EPCP009 booster and EPCP009 protein regimens as tuberculosis vaccine candidates" reports findings on the development of a BCG-based recombinant vaccine against tuberculosis. The study was conducted comprehensively and the results are reliable. However, the manuscript needs amendments:

- The name of the fusion protein, EPCP009, was sometimes mentioned as EP009. I think "EPCP009" can be simplified such as "EPCP".

- Lines 35-36: The sentence should contain "in humans" because TB is caused by different species in animals.

- Lines 44-45: The sentence needs revision because there are promising strategies not including BCG.

- Lines 103-104: It is better to mention that ESAT-6 and CFP-10 are encoded by  esxA  and esxB genes.

- Line 133: The sources of the bacteria should be mentioned.

- Line 142: More information should be provided about the cloned genes, such as GenBank accession numbers, if cloned partially or whole, and the length of the fusion gene.

- Line 150: "electroporation" should be used instead of "electroshock".

- Line 166: How long was the culture induced with IPTG?

- Line 197: Please name the nine cytokines.

- Line 212: The "serum" should be specified.

- All figures should be placed after they are cited in the text and the figure legends should be placed after figure.

- Figure 2: "16s" should be "16S rRNA gene". The expected size of fusion gene and the fusion protein should be given. Meaning of three stars (***) should be specified.

- Previous studies on the BCG prime-boost, such as doi:10.3906/biy-2108-41, should be discussed more.

Author Response

The manuscript entitled "Comparison of the immunogenicity and efficacy of rBCG-EPCP009, BCG prime-EPCP009 booster and EPCP009 protein regimens as tuberculosis vaccine candidates" reports findings on the development of a BCG-based recombinant vaccine against tuberculosis. The study was conducted comprehensively and the results are reliable. However, the manuscript needs amendments:

Thank you very much for reading and reviewing our manuscript, which helps us to improve it to a better scientifical level. We have revised our manuscript, and a lot of changes have been made. We have sent the revised manuscript and a version containing all visible changes.

In the following, points mentioned by the reviews will be discussed:

- The name of the fusion protein, EPCP009, was sometimes mentioned as EP009. I think "EPCP009" can be simplified such as "EPCP".

Response: Thank you for your constructive comments. We have changed the fusion protein as EPCP009. In all of our work, multiple combinations were studied, and “009” was one of the designations.

- Lines 35-36: The sentence should contain "in humans" because TB is caused by different species in animals.

Response: Thank you for your constructive advice. We have added the information in the revised manuscript in lines 35-36.

- Lines 44-45: The sentence needs revision because there are promising strategies not including BCG.

Response: Thank you for your comments. The promising strategies include recombinant BCG vaccines.

- Lines 103-104: It is better to mention that ESAT-6 and CFP-10 are encoded by  esxA  and esxB genes.

Response: Thank you for your constructive advice. We have added the information in the revised manuscript in lines 102-103.

- Line 133: The sources of the bacteria should be mentioned.

Response: Thank you for your advices. We have updated the information in line 132.

- Line 142: More information should be provided about the cloned genes, such as GenBank accession numbers, if cloned partially or whole, and the length of the fusion gene.

Response: Thank you for your advices. We have changed the information in line 143 and the gene sequence is shown in Table S1.

- Line 150: "electroporation" should be used instead of "electroshock".

Response: Thank you for pointing this out. We have changed the information in line 150.

- Line 166: How long was the culture induced with IPTG?

Response: Thank you for pointing this out. We have updated the information in line 166.

- Line 197: Please name the nine cytokines.

Response: Thank you for your advices. We have updated the information in line 199.

- Line 212: The "serum" should be specified.

Response: Thank you for your advices. We have updated the information in line 213.

- All figures should be placed after they are cited in the text and the figure legends should be placed after figure.

Figure 2: "16s" should be "16S rRNA gene". The expected size of fusion gene and the fusion protein should be given. Meaning of three stars (***) should be specified.

Response: Thank you for your comments. We changed the "16s" to "16S rRNA gene" and all figures showed the specific p-values.

- Previous studies on the BCG prime-boost, such as doi:10.3906/biy-2108-41IF: 2.2 Q3 , should be discussed more.

Response: Thank you for your advices. We have updated the information in lines 601-603.

Kanglin Wan

National Institute for Communicable Disease Control and Prevention, Chinese Center for Disease Control and Prevention, Beijing, China

Email address: wankanglin@icdc.cn